

# *In vitro* and *in vivo* comparison of transport media for detecting nasopharyngeal carriage of *Streptococcus pneumoniae*

Anneke Steens[1,2], Natacha Milhano[1,3], Ingeborg S. Aaberge[1] and Didrik F. Vestrheim[1]

[1] Infection Control and Environmental Health, Norwegian Institute of Public Health, Oslo, Norway
[2] Faculty of Medicine, University of Oslo, Oslo, Norway
[3] European Programme for Public Health Microbiology Training (EUPHEM), European Centre for Disease Prevention and Control (ECDC), Stockholm, Sweden

Corresponding author
Anneke Steens, anneke.steens@fhi.no

## ABSTRACT

**Background**. As a standard method for pneumococcal carriage studies, the World Health Organization recommends nasopharyngeal swabs be transported and stored at cool temperatures in a medium containing skim-milk, tryptone, glucose and glycerol (STGG). An enrichment broth used for transport at room temperature in three carriage studies performed in Norway may have a higher sensitivity than STGG. We therefore compared the media *in vitro* and *in vivo*.

**Methods**. For the *in vitro* component, three strains (serotype 4, 19F and 3) were suspended in STGG and enrichment broth. Recovery was compared using latex agglutination, quantification of bacterial loads by real-time PCR of the *lytA* gene, and counting colonies from incubated plates. For the *in vivo* comparison, paired swabs were obtained from 100 children and transported in STGG at cool temperatures or in enrichment broth at room temperature. Carriage was identified by latex agglutination and confirmed by Quellung reaction.

**Results**. *In vitro*, the cycle threshold values obtained by PCR did not differ between the two media ($p = 0.853$) and no clear difference in colony counts was apparent after incubation ($p = 0.593$). *In vivo*, pneumococci were recovered in 46% of swabs transported in STGG and 51% of those transported in enrichment broth (Kappa statistic 0.90, $p = 0.063$).

**Discussion**. Overall, no statistical differences in sensitivity were found between STGG and enrichment broth. Nevertheless, some serotype differences were observed and STGG appeared slightly less sensitive than enrichment broth for detection of nasopharyngeal carriage of pneumococci by culturing. We recommend the continued use of STGG for transport and storage of nasopharyngeal swabs in pneumococcal carriage studies for the benefit of comparability between studies and settings, including more resource-limited settings.

## INTRODUCTION

Monitoring carriage of *Streptococcus pneumoniae* (pneumococci) is important for determining changes after vaccine introduction in national immunisation programmes. To enable comparison of results from different studies and countries, the World Health Organization Pneumococcal Carriage Working Group published a set of standard methods for such studies measuring nasopharyngeal carriage of pneumococci (*Satzke et al., 2013*). A medium containing skim milk powder, tryptone soy broth, glucose and glycerol in distilled water (STGG) is recommended for transport and storage of nasopharyngeal specimens, and transport should be done at cool temperatures (*O'Brien et al., 2001*). Studies using STGG in developed countries have generally revealed prevalences of pneumococcal carriage in children of around 30–50% (*Andrade et al., 2014*; *Van Hoek et al., 2014*; *Desai et al., 2015*). In Norway, several carriage studies have been performed using enrichment broth (beef infusion enriched with 5% horse serum and 3.3% defibrinated horse blood (*Kaltoft et al., 2008*); Statens Serum Institute, Copenhagen, Denmark) for transport at room temperature. Carriage prevalence in those studies was around 80% before and after introduction of the 7-valent pneumococcal conjugate vaccine (PCV), and 62% two years after switching to the 13-valent PCV (*Steens et al., 2015*). Although different factors may contribute to this high prevalence, such as the percentage of children in day-care (>90% (*Statistics Norway, 2010*)) and the low use of antibiotics in Norway (*Garcia-Rodriguez & Fresnadillo Martinez, 2002*; *Norwegian Veterinary Institute, 2014*), results suggest that enrichment broth transported at room temperature may be more sensitive for detection of carriage than STGG transported at cool temperatures.

We therefore compared (1) *in vitro* recovery from serial dilutions in STGG and enrichment broth and (2) *in vivo* detection of nasopharyngeal carriage of pneumococci from swabs that were transported and stored in STGG at cool temperatures or in enrichment broth at room temperature.

## MATERIALS AND METHODS

In the *in vitro* component of the study, we compared recovery rates of pneumococci from serial dilutions that had been stored at different temperatures and media using (I) culturing, (II) a commercial latex agglutination kit and (III) quantitative real-time PCR (qPCR). In the *in vivo* component, we compared carriage using paired swabs taken from children attending day-care centres and transported in STGG or enrichment broth. In the second part, we used methods (I) and (II) for detection of pneumococci. See Fig. 1 for a schematic overview of the procedures. Note that we followed the recommended conditions for transport and storage for each medium (wet ice/cool box for STGG and freezing (*in vivo* only), room temperature and immediate processing for enrichment broth).

### *In vitro* comparison

Three strains belonging to different serotypes were used as pneumococcal samples; two reference strains (ATCC49619—serotype 19F, and TIGR4—serotype 4), and a strain belonging to serotype 3 obtained from the 2013 sample of a previous Norwegian carriage study (*Steens et al., 2015*). Colonies from each serotype were suspended in Todd-Hewitt

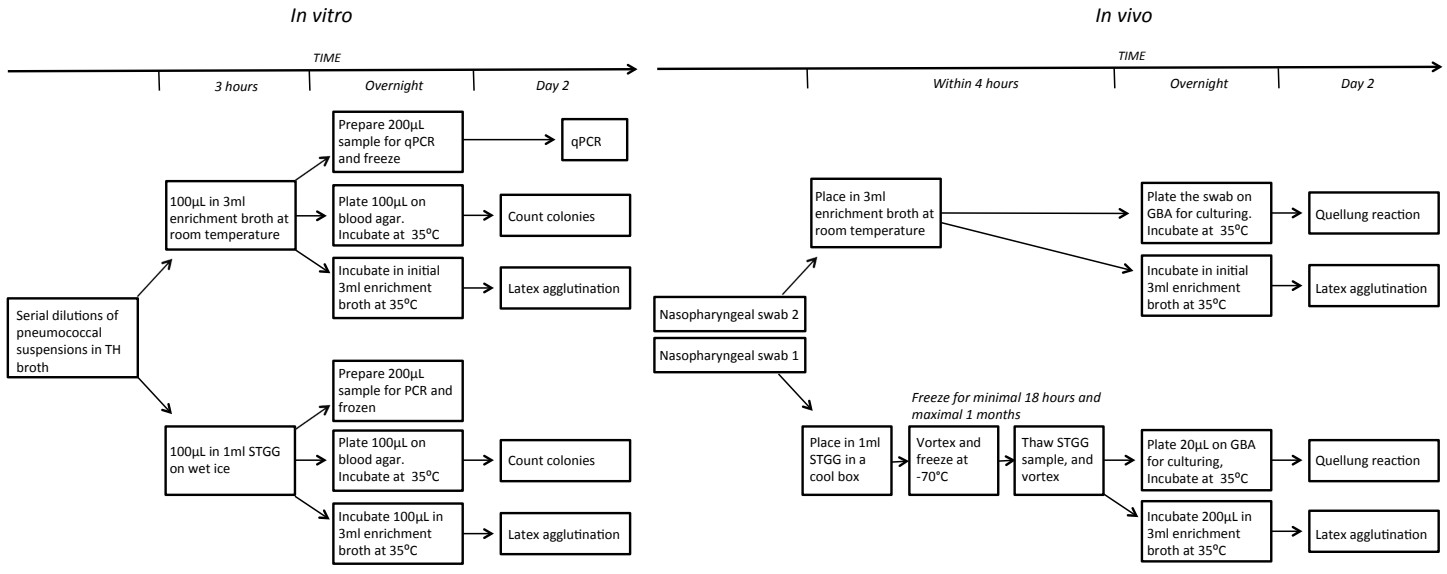

**Figure 1** **Schematic overview of the experimental designs.** (A) *in vitro*. (B) *in vivo*. TH, Todd-Hewitt; STGG, skim milk, tryptone, glucose and glycerol; GBA, gentamycin-blood-agar.

(TH) broth at a concentration of 0.5 McFarland in serial 1:10 dilutions of $10^{-2}$ to $10^{-5}$ (minimal concentration for which recovery of pneumococcal DNA was possible; see Supplemental Information 3). A volume of 100 μL of serotype dilution in TH broth was added to each set of transport media (either 1 ml of STGG or 3 ml of enrichment broth) to prepare the pneumococcal samples. The samples were left for 3 h on wet ice (STGG) or room temperature (enrichment broth). Subsequently, 100 μL of the samples was plated on Columbia horse blood agar plates. Furthermore, 100 μL of the STGG samples was added to 3ml fresh enrichment broth. All was done in triplicate. Plates and tubes (broth sample made from the STGG samples and the initial enrichment broth samples) were incubated overnight at 35 °C with 5% $CO_2$.

Pneumococci were identified by latex agglutination (Pneumotest-Latex kit; Statens Serum Institut, Denmark; *Slotved et al., 2004*) from the incubated broths. Quantification of the bacterial loads was performed by qPCR (see below for details) and counting of the colony forming units (CFU) from the incubated plates.

### DNA extraction and amplification by qPCR

From each sample 200 μL was boiled for 10 min and DNA was extracted by QIAamp DNA Mini QIAcube kit (Qiagen, Inc., Valencia, CA, US) according to the manufacturer's recommendations. A qPCR assay for the detection of the autolysin-encoding gene (*lytA*) was then performed as described before by *Carvalho et al. (2007)*. Briefly, 25 μL reaction volume composed of TaqMan Fast Universal PCR Master Mix 2x, 200 nM of each primer and probe, 10x Exo IPC-mix, 50x Exo IPC DNA and 2 μL of DNA was run at 50 °C for 2 min, denaturation at 95 °C for 10 min, followed by 40 amplification cycles of 95 °C for 15 s and 60 °C for 1 min. Samples were considered negative if cycle thresholds (Ct)

exceeded 40. A positive (ATCC49619) and a non-template control (sterile water) were included in each run, along with extraction controls.

### In vivo comparison

This comparison was performed as part of a larger carriage study (sample taken in 2015 (*Steens et al., 2016*)). The study was conducted in accordance with principles of the Declaration of Helsinki, and approved by the Regional Committee for Medical Research Ethics, South-Eastern Norway (reference number: 2014/2046). Parents/guardians of the children gave written informed consent before including their child in the study. The study design resembles the design used in previous Norwegian carriage studies (*Steens et al., 2015*).

Two flocked nylon nasopharyngeal swabs (E-swabsTM, Copan, Italy) taken from the same nostril were collected from 100 children aged 1–6 years according to standard procedures. The first swab was placed in 1 ml STGG which was subsequently stored in a cool box and the second swab was stored and transported in 3 ml enrichment broth at room temperature. The specimens were processed within 4 h of sampling. The STGG samples were vortexed at high speed and frozen at $-70\,°C$, following the recommendations of WHO (*Satzke et al., 2013*). Within one month but earliest 18 h after initial freezing, the STGG samples were processed further: after being thawed and vortexed, 200 μl was added to fresh enrichment broth and 20 μl was plated on gentamycin-blood-agar (GBA). The swabs from the enrichment broth samples were plated on GBA within 4 h of sampling and the swab was re-inserted into the enrichment broth. All broths and GBA plates were incubated overnight at $35\,°C$, with 5% $CO_2$.

Pneumococci were identified by latex agglutination from incubated broths. Confirmation and factor typing were performed by Quellung reaction. All morphologically different pneumococcal colonies per plate were typed. In cases where the latex agglutination was positive but no colonies were found on plates after incubation overnight, samples were re-cultured by plating one drop of the incubated broth and incubating this for another night before further analysis.

### Data analysis

For the *in vitro* analyses, Ct values and CFU counts (after a logarithmic transformation; logCFU) from the two media were compared by linear regression. We used 11 mutually exclusive dummy variables identifying different dilution-serotype combinations, which enabled us to simultaneously run the comparison for all serotypes. Additionally, the analyses were conducted separately per serotype.

Agreement in the *in vivo* comparison was determined using the kappa statistic (*Landis & Koch, 1977*) and the exact McNemar's probability test for paired data. Data were analysed in Stata 14.0 and GraphPad Prism 5.

## RESULTS

### In vitro results

The latex agglutination test was positive for both media for all serotype dilutions tested, with the exception of serotype 3 at a dilution of $10^{-5}$ in STGG, where no pneumococci were detected.

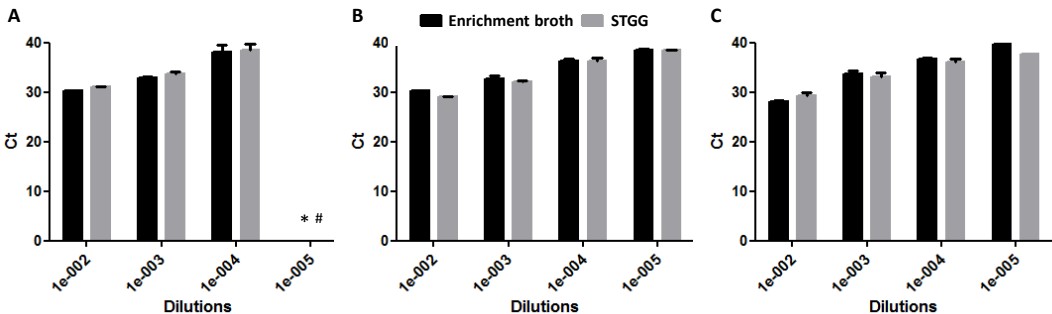

**Figure 2** DNA quantification (Ct values) at different dilutions of enrichment broth and STGG. (A) serotype 19F*; (B) serotype 4; (C) serotype 3. * At a dilution of $10^{-5}$, one of the triplicates had a cycle threshold (Ct) of 39.4, while the other two were above 40. This sample was therefore considered negative. # Ct above 40.

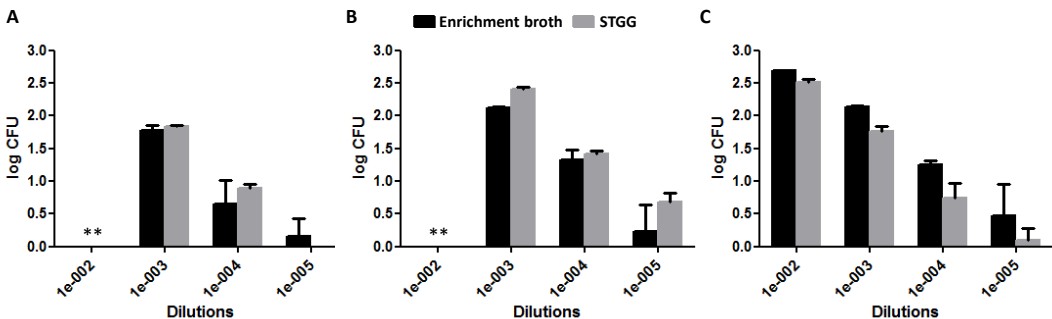

**Figure 3** Quantification of bacterial load (log counts of colony forming units [CFU]) at different dilutions of enrichment broth and STGG. (A) serotype 19F; (B) serotype 4; (C) serotype 3. * Uncountable.

The quantification of DNA by Ct value is presented in Fig. 2. The Ct values overall did not differ significantly between STGG and enrichment broth samples ($p = 0.853$), though for serotype 4, the Ct values were significantly lower for STGG samples compared to enrichment broth samples ($p = 0.007$). After culturing, no clear overall difference was found in the logCFU ($p = 0.593$) but significant differences were observed for serotype 4 (Fig. 3B; $p = 0.008$; more colonies on plates incubated with STGG samples) and serotype 3 (Fig. 3C; $p = 0.001$; more colonies on plates incubated with enrichment broth samples).

### *In vivo* results

Forty-six percent of the swabs transported and stored in STGG and 51% of those transported in enrichment broth were positive for pneumococci. This resulted in a Kappa statistic for carriage of 0.90 for the paired swabs (Table 1), indicating a trend towards higher sensitivity after transportation in enrichment broth compared to STGG ($p = 0.0625$). If re-cultured samples were excluded, carriage was 44% for the samples transported and stored in STGG and 48% for those transported in enrichment broth (Kappa = 0.92; $p = 0.125$). For each child for whom both swabs were positive, the same serotype was obtained. In one child, carriage of two serotypes was found in the enrichment broth sample, while one serotype was found in the STGG sample. However, the plate incubated with the enrichment broth

**Table 1  Pneumococcal carriage determined by culturing paired nasopharyngeal swabs stored in STGG or enrichment broth.**

|  |  | Enrichment broth | | |
|---|---|---|---|---|
|  |  | Positive | Negative | *Total* |
| **STGG** | **Positive** | 46 | 0 | 46 |
|  | **Negative** | 5 | 49 | 54 |
|  | *Total* | 51 | 49 | 100 |

Notes.
STGG,  skim milk, tryptone, glucose and glycerol.

sample had only one colony of the serotype that was missed in the STGG sample (serotype 3) and the agglutination test was negative for this serotype, indicating presence at a very low concentration.

## DISCUSSION

Overall, no statistical differences in sensitivity were found between STGG stored and transported at cool temperatures and enrichment broth transported at room temperature. Nevertheless, some serotype differences were observed as well as a trend towards higher sensitivity for detection of pneumococcal carriage after transportation in enrichment broth compared to STGG.

There are several possible reasons for these differences. *In vitro*, a pure dilution of one serotype was used in place of a nasopharyngeal swab. *In vivo* the swab contained different respiratory bacteria and viruses, and was covered with mucus and cellular debris. The presence of other bacteria places pneumococci in competition for nutrients needed for growth and reproduction. These nutrients are available in higher concentration in enrichment broth than in STGG, which may explain the small difference in sensitivity observed between the *in vitro* and *in vivo* settings. In addition to the difference in available nutrients in STGG and enrichment broth, the fact that STGG samples were kept in a cool box during transport while the enrichment broth samples were kept at room temperature, may have caused the small non-significant difference in carriage prevalence. The original carriage study presenting the use of enrichment broth as transport medium used cool transport conditions (*Kaltoft et al., 2008*) but three previous Norwegian carriage studies used room temperature (*Steens et al., 2015*). The present study is not designed to differentiate between the effect of media and temperature, but while pneumococci are thought to thrive better at warmer temperatures, we did not find an overall difference between the methods in bacterial load or DNA quantity *in vitro*. Whether the relative abundance of other respiratory bacteria/the microbiome may have changed and would no longer be representative of the nasopharyngeal tract after storage at room temperature should be investigated were room temperature to be used in such studies. Still, for serotype 3, the bacterial load indicated a higher sensitivity of enrichment broth compared to STGG. The serotype 3 isolate was obtained from a previous carriage study. For serotype 19F and 4, reference strains were used. The origin of the isolates (reference strains versus carriage isolate) may have induced different bacterial growth characteristics. Further, the capsule structure differs between

serotypes (serotype 3 being very mucoid). The low number of serotypes tested and the difference in origin between serotypes are limitations to the *in vitro* part of this study.

The carriage prevalence found in this study is lower than observed previously in Norway (*Steens et al., 2015*), and more similar to what has been seen in other developed countries (*Van Hoek et al., 2014*; *Desai et al., 2015*; *Andrade et al., 2014*). The methods used for swab collection, transport and incubation in enrichment broth and culturing were unchanged from former studies, indicating a real difference in carriage prevalence that may have resulted from vaccination.

Advantages of molecular based techniques compared to culture techniques include the fact that viable organisms are not required, the original composition of the nasopharyngeal specimen is preserved, and detailed quantification and characterization of the pneumococci within a sample are possible, depending on the methods used (*Satzke et al., 2013*). Furthermore, the sensitivity of molecular methods for detection of multiple co-colonising serotypes has been shown to be higher than conventional methods (*Saha et al., 2015*). Nevertheless, isolation of strains enables further characterization such as antimicrobial susceptibility testing and sequence typing, and should not be replaced by molecular methods alone, despite its high sensitivity. The additional incubation step and use of latex agglutination appears to be of value for identification and isolation of pneumococci from multiple carriage (*Kaltoft et al., 2008*). The PneuCarriage Project concluded that microarray with a culture amplification step has the highest sensitivity for determining carriage (*Satzke et al., 2015*).

Finally, STGG is cheap, easy to make and can be stored longer than enrichment broth, thus enabling comparability between studies and settings, including more resource-limited settings. Furthermore, STGG transported and stored at cool temperatures enables studies to investigate the microbiome (*Grijalva et al., 2014*; *Turner et al., 2011*), whereas enrichment broth may selectively stimulate growth of pneumococci. Therefore, even though STGG appeared slightly less sensitive than enrichment broth for detection of nasopharyngeal carriage of pneumococci by culturing, we recommend the continued use of STGG for transport and storage of nasopharyngeal swabs at cool temperatures in future carriage studies.

## ACKNOWLEDGEMENTS

We thank the children and parents for participating and day-care centres workers for their support. We also thank Anne Ramstad Alme, Gunnhild Rødal, Lene Haakensen, Anne Witsø and Martha Langedok Bjørnstad for the laboratory analyses, Ingvild Essén and Kristine Hartmark for the collection of nasopharyngeal swabs, and Richard White for statistical advice.

### Funding

The authors received no funding for this work.

## Competing Interests

The authors declare there are no competing interests.

## Author Contributions

- Anneke Steens and Natacha Milhano conceived and designed the experiments, performed the experiments, analyzed the data, contributed reagents/materials/analysis tools, wrote the paper, prepared figures and/or tables, reviewed drafts of the paper.
- Ingeborg S. Aaberge conceived and designed the experiments, contributed reagents/materials/analysis tools, reviewed drafts of the paper.
- Didrik F. Vestrheim conceived and designed the experiments, performed the experiments, contributed reagents/materials/analysis tools, reviewed drafts of the paper.

## Human Ethics

The following information was supplied relating to ethical approvals (i.e., approving body and any reference numbers):

Regional Committee for Medical Research Ethics, South-Eastern Norway (REK sør-øst B) Approval number: 2014/2046.

## Data Availability

The raw data has been supplied as a Supplementary File.

## Supplemental Information

Supplemental information for this article can be found online at http://dx.doi.org/10.7717/peerj.2449#supplemental-information.

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
