# Peer review of "In vitro and in vivo comparison of transport media for detecting nasopharyngeal carriage of Streptococcus pneumoniae"

_PeerJ, doi:10.7717/peerj.2449_

## Round 0.1 · original submission · Minor Revisions

· Academic Editor

Minor Revisions

Both reviewers were positive about this manuscript but suggest a number of text and figure modifications. No additional experiments are needed.

·

Basic reporting

No Comments

Experimental design

1. I find that the big difference of around 20C for 4 hours when stored either on wet ice or at room temperature is very problematic. Pneumococcal isolates can grow very fast at optimal conditions such as in enrichment broth. So is it not a problem that the enrichment media have around four hours of better pneumococcal growth conditions than the STGG on wet ice? In lines 115 to 125, the authors describe the handling of both transport media, and to me there seems to be a very vast difference in the conditions between both transport media. I think that this big difference in the handling of transport media makes it very difficult to compare the two broths. I furthermore think the authors have to mention and discuss this in the discussion section.
2. Why did the authors not also try to handle either the STGG or the enrichment broth in an identical collection procedure with transportation at either 4C or 20C? The authors find a slightly non-significant lower sensitivity with the STGG compared to the enrichment broth. What do the authors think would have happened if the STGG media were stored at room temperature when transported? The STGG media is a good growth media for pneumococci. Is it possible that this slight difference would have disappeared? If you had stored the enrichment broth on wet ice, what would have happened then?
3. I think it is good to learn that enrichment broth transported at room temperature for 4 hours does not change the carriage rate significantly compared to STGG transported on ice. This will probably be the same for STGG if you have conditions making it problematic to transport it on wet ice.
4. It is difficult to see from which study the samples were collected. Are the carriage study described in “abstract ISPPD-0116 (2016) the same as the carriage study described in ref 12, or are they two different studies? I think this should be specified more in the text.
5. Regarding Abstract ISPPD-0116, 2016, I think it should be Abstract ISPPD-370, 2016. Furthermore it should be added to the reference list with author names and title.
6. In the study, the authors use real-time PCR for detection of lytA; however, recently a study showed this PCR to misidentify S. pneumoniae (Simões et al., lytA-based identification methods can misidentify Streptococcus pneumoniae, Diagnostic Microbiology and Infectious Disease (2016), doi:10.1016/j.diagmicrobio.2016.03.018). How much do the authors believe that this will affect their results?
7. As I understand it, the authors used real-time PCR on both broths in the in vitro study, while in the in vivo study, real-time PCR was used only on the enrichment broth. Does figure 1 describe the in vivo study? I suggest that the authors make a figure 1 where both the in vitro and in vivo setups are described. This will make it clearer how the two experimental designs were performed.

Validity of the findings

No Comments

Additional comments

In this study, the authors compare two transport media, the STGG media and an enrichment broth. You collect two swab samples from each participant, and one (STGG) is placed directly in wet ice, while the other (enrichment broth) is placed in room temperature. The samples were processed within 4 hours. In general, I think that the two different procedures for swab transportation represent two different purposes. The STGG broth and procedure are made for collecting swab samples, to preserve the microbiome situation from when the sample are collected. The enrichment broth and procedure seem to me only to favour the pneumococcal detection and not the microbiome at collection. I therefore find it very positive that the STGG gives a very similar pneumococcal output compared to the enrichment broth procedure.

·

Basic reporting

This article satisfies the criteria for basic reporting, demonstrating how the work fits into the broader field of knowledge. However, I would suggest restricting the comparison of carriage prevalence (in the Introduction) to developed countries (e.g. references [3] and [5]). Carriage is considerably higher than 50% in African countries (refs [8] and [9]) and other disadvantaged populations (e.g. Indigenous children), and there are significant subgroup differences in Alaska (rural vs urban, refs [4] and [7]) and Fiji (ref [6]).
The structure is standard and the number of figures and tables appropriate. The English is understandable but suggestions for improvement and 'tightening' of the text have been made (see attached Word document). The last two sentences under Conclusions should rather appear in the Discussion. Some details are missing from some of the references. Figures 2 and 3 use the term ‘serum broth’ instead of ‘enrichment broth’.

Experimental design

No comments, except that I do not understand the use of dummy variables for the linear regression.

Validity of the findings

The findings are valid.

Additional comments

Figure 1 is difficult to follow; it appears that samples in STGG did not undergo PCR. The figure should better match the description in the Methods section (see Fig1 attached), and amounts used (e.g. 200 µL, 20 µL) added to aid interpretation.

---

## Round 0.2 · accepted · Accept

· Academic Editor

Accept

All reviewer comments have been dealt with appropriately.